# Development of CNT-Based Nanocomposites with Ohmic Heating Capability towards Self-Healing Applications in Extrusion-Based 3D Printing Technologies

Niki Loura [1], Eleni Gkartzou [1], Aikaterini-Flora Trompeta [1], Georgios Konstantopoulos [1], Panagiotis A. Klonos [2], Apostolos Kyritsis [2] and Costas A. Charitidis [1,*]

[1] Research Lab of Advanced, Composite, Nano-Materials and Nanotechnology (R-NanoLab), School of Chemical Engineering, National Technical University of Athens, 9 Heroon Polytechniou, GR-15780 Athens, Greece; nikil4@hotmail.com (N.L.); egartzou@chemeng.ntua.gr (E.G.); ktrompeta@chemeng.ntua.gr (A.-F.T.); gkonstanto@chemeng.ntua.gr (G.K.)

[2] Dielectrics Group, Physics Department, School of Applied Mathematical and Physics Science, National Technical University of Athens, 9 Heroon Polytechniou, GR-15780 Athens, Greece; panos48al@gmail.com (P.A.K.); akyrits@central.ntua.gr (A.K.)

[*] Correspondence: charitidis@chemeng.ntua.gr

**Abstract:** In the present study, a series of carbon-based nanocomposites based on recycled thermoplastic polyurethane (TPU) matrix and MWCNT fillers synthesized in a laboratory environment were prepared at various loadings and assessed in terms of their functional thermal, dielectric, and rheological properties, as well as their ohmic heating capability, for self-healing applications in extrusion-based 3D printing technologies. The synthesis of nanomaterials focused on the production of two different types of carbon nanotubes (CNTs) via the chemical vapor deposition (CVD) method. A comparative assessment and benchmarking were conducted with nanocomposite filaments obtained from commercial nanomaterials and masterbatches with MWCNTs. For all the polymer nanocomposites, samples were prepared at additive contents up to 15 wt.% and filament feedstock was produced via the melt-extrusion process for 3D printing; these were previously characterized by rheological tests. The measurements of thermal and electrical conductivity resulted in a selected composition with promising ohmic heating capability. As a preliminary assessment of the self-healing ability of the above samples, artificial cracks were introduced on the surface of the samples and SEM analysis took place at the crack location before and after applying voltage as a measure of the effectiveness of the material remelting due to the Joule effect. Results indicate a promising material response with a partial restoration of artificial cracks.

**Keywords:** carbon nanotubes; nanocomposites; extrusion; 3D printing; rheology; electrical conductivity; thermal conductivity; self-healing; recycling; TPU

## 1. Introduction

The ability to inherently repair damage without external intervention has gained significant interest in the scientific community, resulting in an emerging class of smart materials with higher endurance limits and prolonged service-life through self-healing functionalities [1–4]. Functional polymers and fillers present several advantages for producing self-healing nanocomposites due to their ease of processability and wide range of combinations available in the material design space, enabling simultaneous tailoring of application-specific properties (e.g., mechanical, thermal, and electrical properties) as well as responses to different external stimuli, such as heat, light, or electricity, to activate self-healing mechanisms. For instance, the incorporation of conductive nanofillers in thermoplastic matrices has been reported as a common approach for introducing self-healing functionality, as the formation of interconnected pathways of randomly dispersed

filler material introduces a percolation network that provides resistive heat generation upon passage of an electric current [5–7]. This so-called Joule heating effect triggers the intrinsic self-healing ability of the thermoplastic matrix, raising the temperature of the damaged area in a localized way due to the change in the direction of the electric current at the tips of micro-cracks and cracks and providing the necessary activation energy to achieve polymer chain diffusion into the damaged zone [5]. For polymers, the mobility of macromolecular chains occurs at temperatures above their glass transition temperature ($T_g$); therefore, temperature plays an important role in the self-healing process [8]. In this context, the Joule effect is an ideal candidate as a self-healing stimulus for conductive polymer nanocomposites [5,9,10].

Carbon-based nanomaterials offer the possibility to combine polymer properties with several of their unique features, such as high mechanical strength and electrical and thermal conductivity [11]. Carbon nanotubes (CNTs), besides their high electrical and thermal properties, are widely known for their use as fillers in preparing functional composites [12–14]. For the successful incorporation of CNTs in polymer matrices, effective production methods for large-scale control of CNT properties, with high yield and purity are required. Among the available methods for CNT production, chemical vapor deposition (CVD) is a simple and cost-efficient technique for synthesizing CNTs at a low temperature and ambient pressure. In terms of controlling structures and growth parameters in CNTs, CVD is the optimal solution [15,16]. During the CVD processes, the catalyst is the most fundamental variable. Depending on the catalytic mechanism and catalyst type, the CVD system is categorized as a cCVD (catalytic CVD) system with a floating [17–19] or supported catalyst [20,21].

The dispersion and distribution of nanoparticles in the polymer matrix is crucial in obtaining new material systems with synergistic effects arising from the constituent properties for the optimization of isotropic bulk materials. Regarding the methods of preparation of nanocomposite materials, filler incorporation and polymer bulk modification through compounding and melt-extrusion are established processing routes compatible with conventional polymer processing facilities, enabling the production of high-performance polymer nanocomposites on a larger scale [22,23]. In order to improve filler dispersion, concentrated mixtures of additives (masterbatches) are frequently employed as a preparatory step with the aim of achieving a higher degree of control over the amount and type of additives in the polymer matrix and overall consistent quality [24].

Among the thermoplastic polymer matrixes, thermoplastic polyurethanes (TPU) have received special attention because of their favorable properties such as stability, flexibility, good abrasion resistance, chemical and corrosion resistance, and mechanical properties [25,26]. They can also be considered as functional polymers, presenting a promising path to heal microcracks due to autonomous healing features [27]. CNT-based self-healing polymer composites have also been investigated in several studies [28–32]. Gu et al. [33], through the incorporation of different types of CNTs in a TPU matrix, proved that the self-healing mechanism is dependent on the interactions between the fillers and the matrix. Luan et al. [34], reported the healing performance of TPU composites, which were filled with graphene CNTs, under microwave radiation.

Fused filament fabrication (FFF) is a low-cost 3D printing technology with an ever-increasing range of feedstock materials, suitable for a wide range of 3D printing applications. It is an additive manufacturing (AM) technology that involves the fabrication of physical models via layer-by-layer deposition of extruded filaments using computer-aided design (CAD) data [35]. Prior to FFF processing, raw materials and masterbatches are subjected to thermal processing and circular cross-sectional profile extrusion in order to produce FFF filaments. An important filament quality characteristic to obtain is a constant diameter within the $1.75 \pm 0.05$ mm range [36]. In the FFF process, fundamental importance lies in the rheological properties of the melt, as they can considerably affect the printing parameters [37–39]. In terms of FFF material development, another significant challenge facing the scientific community is the development of sustainable nanocomposites for functional

applications. The use of recycled filaments or filaments with a recycled polymeric matrix can contribute significantly to a reduction in disposal and production costs, increasing the sustainability of the final product [40].

Additionally, the combination of 3D printing technologies with material feedstocks with embedded functionalities is considered highly promising for the production of functional parts with extended service lives [41]. Several researchers have shown that it is possible to fabricate 3D-printed sensors using functionalized nanocomposites based on CNTs and TPU [42–44]. Niu, Yang et al. [45] showed that 3D-printed CNT/TPU composites could offer both EMI-shielding performance and mechanical properties in next generation electronic devices. Wu, Zou et al. [46] proves the possibility of 3D printing self-healable strain gauges and humidity sensors using CS/CNT nanocomposites because of their high stretchability and conductivity.

This work systemically investigates the use of nanocomposites with recycled polymer matrixes and MWCNT fillers synthesized in a laboratory environment as self-healing conductive filaments for 3D printing. Specifically, in this research, CNT-derivatives in different loadings in a recycled TPU matrix were assessed in terms of their rheological properties and printability, as well as their functional electrical and thermal properties related to self-healing capability. A comparative assessment and benchmarking were also conducted with nanocomposite filaments obtained from a commercial masterbatch with MWCNTs.

## 2. Materials and Methods

### 2.1. Materials

The TPU polymer matrix selected for the study was TPU Elastollan® 1164 D by BASF Polyurethanes GmbH (Ludwigshafen, Germany) (reference name: rTPU), originating from post-industrial scrap from injection moulding processes that was shredded and subjected to one reprocessing cycle. rTPU pellets were dried overnight at 60 °C before processing. Nanocomposites for the selected polymer matrix were prepared with three types of carbon-based nanofillers, namely: (i) MWCNTs produced in-house via the supported catalyst CVD method (reference name: MWCNTs_csCVD); (ii) in-house produced byproduct from the floating catalyst CVD synthesis of MWCNTs (reference name: MWCNTs_cfCVD_bp); and (iii) commercial MWCNTs (reference name: MWCNTs_com) supplied from Hongwu International Group Ltd. (Hong Kong, China), with average diameter of 20 nm and average length of 10 μm (average aspect ratio of 500). For the preparation of rTPU masterbatches, DMF solvent was employed for the solution mixing processes and was purchased from Sigma Aldrich. In addition, benchmarking formulations were prepared through the dilution of commercial masterbatch 10%CNTs/TPU PLASTICYL™ TPU1001 (reference name: mTPU_1001) purchased from Nanocyl SA, incorporating NC7000 MWCNTs (average aspect ratio of 157).

### 2.2. Synthesis of Nanomaterials

#### 2.2.1. Synthesis of MWCNTs

The production of MWCNTs_csCVD was carried out through the supported catalyst method based on the already optimized method proposed by Trompeta et al. [47]. The experimental setup consisted of the CVD reactor, which is a horizontal cylindrical furnace containing a quartz tube in which silicon wafer substrates are placed, carrying sieved iron catalyst adsorbed into zeolite. Acetylene was chosen as the carbon source at a volumetric flow rate of ~60 mL/min, and argon was chosen as the carrier gas at a volumetric flow rate of ~200 mL/min. The experiment lasted for 4 h at a reaction temperature of 700 °C, resulting in 7.5 g of MWCNTs produced per batch. The above procedure was repeated four times in order to obtain the necessary amount of MWCNTs for the production of the nanocomposites.

### 2.2.2. Synthesis of CVD Byproduct

Another type of CVD byproduct was chosen to be tested: a carbonaceous material collected during the CVD floating catalysis process (MWCNTs_cfCVD_bp). It consists of a mixture of nanotubes, carbon nanofibers, and amorphous carbons that grow on the metallic walls of the reactor during a conventional floating catalysis reaction [48]. During the reaction, ferrocene acted as a catalyst, while camphor and ethanol were used as precursors. The reaction was carried out at 850 °C under a constant flow of nitrogen equal to 300 mL/min. Because the catalyst was now contained in the gaseous mixture, it was possible to develop a variety of carbon nanostructures, which were deposited on the metallic walls of the reactor. On average, in each reaction, ~7 g of byproduct was collected. The above procedure was repeated four times in order to obtain the necessary amount for the production of the nanocomposites.

### 2.3. Preparation of Nanocomposites

### 2.3.1. Masterbatch Preparation

In order to facilitate nano-filler dispersion, high-content masterbatches (MB) were produced at 20 wt.% nanofiller loadings. rTPU masterbatch preparation was conducted via the solution mixing method, using dimethylformamide (DMF) for matrix dissolution and nanofiller dispersion [49–51]. More specifically, the experimental method was based on the initial dissolution of rTPU and each additive in separate dispersions in DMF and subsequent mixing of the two parts under stirring to ensure the fine dispersion of nanofillers; this was done while retaining a high temperature of about 100–120 °C to ensure control over the solvent evaporation. Upon extrusion, the mixture was converted into pellet form through a pelletizer and dried overnight prior to processing. Commercial masterbatch mTPU_1001 was also dried overnight and used as received for the preparation of benchmark samples.

### 2.3.2. Twin-Screw Extrusion

After the 20 wt.% masterbatches were produced from the selected nanofillers, the individual contents of 1–5–10–15 wt.% were prepared by dilution with the respective polymer matrix in a twin-screw extruder (Table 1), starting from the lowest content to the highest (Figure 1). Regarding the concentration of the additives, additive contents higher than 15 wt.% led to rheological constraints during filament production, as constant diameter was difficult to achieve due to melt flow instability associated with the viscosity increase. In addition, benchmarking samples with 1, 5, and 10 wt.% MWCNT contents were prepared from the commercially available masterbatch. The extruder unit employed for compounding and filament production was a Thermo Scientific™ Process 11 Twin-Screw Extruder (Dreieich, Germany). In order to obtain a close-tolerance extrudate of 1.75 mm diameter at the die, a melt pump system was positioned in line, downstream of the extruder screw and barrel, and the extrudate diameter was monitored through a triaxial laser in-line measuring unit.

**Table 1.** The extruder spinning profile for TPU nanocomposite processing.

| Nanocomposites | Feeder Speed (rpm) | Screw Speed (rpm) | Zone 2 (°C) | Zone 3 (°C) | Zone 4 (°C) | Zone 5 (°C) | Zone 6 (°C) | Zone 7 (°C) | Zone 8 (°C) | Die (°C) |
|---|---|---|---|---|---|---|---|---|---|---|
| rTPU, mTPU_1001 | 20–30 | 300 | 100 | 180 | 205 | 210 | 210 | 210 | 210 | 210 |

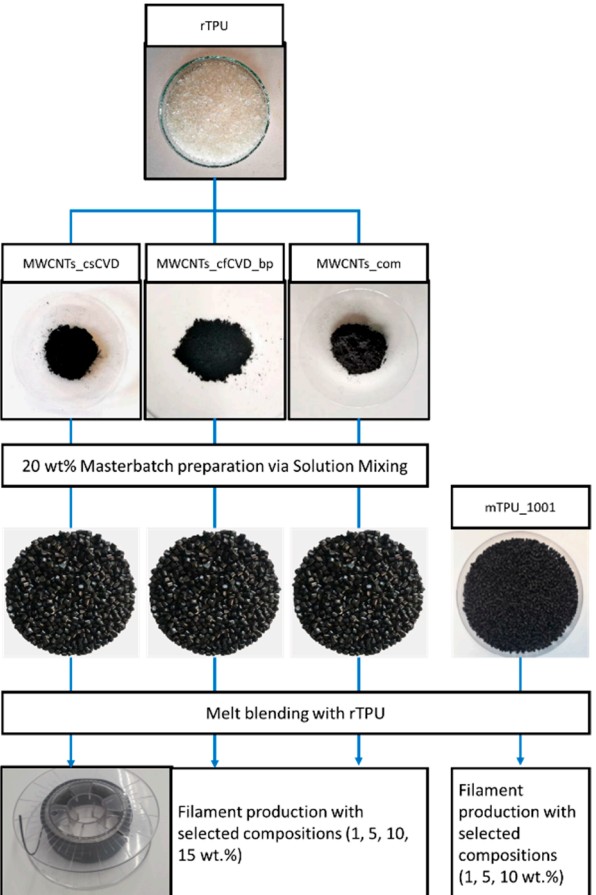

**Figure 1.** A schematic diagram following from the raw materials to the filament production.

A list of compositions prepared and reference names of samples produced is included in Table 2.

**Table 2.** The studied nanocomposites with their contents by weight of their additives.

| Matrix | Additive/MB | Contents by Weight % of Nanofiller | Reference Name (X = 1, 5, 10, 15; Y = 1, 5, 10) |
|---|---|---|---|
| rTPU | MWCNTs_csCVD MWCNTs_cfCVD_bp MWCNTs_com mTPU_1001 | 1–5–10–15% 1–5–10% * | rTPU/MWCNTs_csCVD_Xwt rTPU/MWCNTs_cfCVD_bp_Xwt rTPU/MWCNTs_com_Xwt rTPU/mTPU_1001_Ywt |

* mTPU_1001_10wt was the maximum concentration (10 wt.%) of the commercial masterbatch pellets (mTPU_1001); thus, filament was produced from the as-received commercial pellets without blending with rTPU.

### 2.4. Preparation of Samples for Characterization

For the rheological and electrical conductivity characterization, circular cross-section specimens of 25 mm diameter and 1.5 mm thickness were prepared by injection moulding in an Xplore micro-injection Moulder IM 12 system ('reference name'_IM). The process parameters are shown in Table 3.

**Table 3.** Injection moulding parameters for the nanocomposites.

| Nanocomposites | $T_{mould}$ (°C) | $T_{melt}$ (°C) | P (bar) |
|---|---|---|---|
| rTPU, mTPU_1001 | 65 | 215–225 | 10–12 |

In addition, 3D-printed specimens were prepared for electrical and thermal conductivity characterization ('reference name'_3D). Nanocomposite filaments were processed in a PRUSA i3 MK2 direct drive FFF system with a 0.4 mm diameter E3D V6 stainless steel nozzle. Duplicates were printed for each material in a circular shape of 25 mm diameter and 5 mm thickness. The dimensions of the specimens were based on the limits set by the thermal conductivity evaluation layout. The 3D designs were prepared using Fusion360 CAD software by Autodesk (v. 14.13.0.1525), and the printing settings and control were performed using the PrusaSlicer slicing software (v2.3). The selected FFF parameters for specimen fabrication are presented in Table 4. Finally, the ohmic heating behavior of the developed compositions was evaluated in segments of produced filaments with 1.75 mm diameters.

**Table 4.** 3D printing process parameters applied for the studied nanocomposites.

| 3D Printing Conditions | Nanocomposites rTPU, mTPU_1001 |
|---|---|
| Temperature bed (°C) | 60 |
| Nozzle temperature (°C) | 240 |
| Printing speed (mm/s) | 30 |
| Layer thickness (mm) | 0.2 |
| Fill density (%) | 100 |

*2.5. Characterization Methods*

2.5.1. Characterization of Nanomaterials

A Hitachi TM3030 tabletop microscope (QUANTAX 70), equipped with a diaphragm pump and a turbomolecular pump and operated in secondary electrons mode with an acceleration voltage of 10 and 15 keV in a vacuum of $10^{-5}$ Pa, was employed for scanning electron microscopy (SEM) in order to analyze the morphology of the synthesized nanomaterials. X-ray diffraction (XRD) measurements were performed with a Bruker Advance D8 diffractometer (Cu K$\alpha$, $\lambda$ = 1.5418 Å) to analyze the crystallinity phase present in the samples from 8° to 70°. Micro-Raman analysis was performed in backscattering configuration on a Renishaw inVia Reflex microscope using a diode laser ($\lambda$ = 532.0 nm) as excitation source with the scope to analyze the quality of the produced materials at wavelengths from 1000–4000 cm$^{-1}$.

2.5.2. Rheology Tests

The dynamic shear rheology of the materials was measured using a Thermo Scientific™ HAAKE™ MARS™ rheometer. Test specimens were dried in a vacuum oven at 80 °C for 10 h before the test. A gap of 1 mm between the parallel plates was used for all tests. All frequency and amplitude sweep measurements were performed at 240 °C. To maintain a linear response, the linear viscoelastic region (LVR) was first determined, and, subsequently, frequency sweep tests were conducted on test specimens in the $\omega$ range of 0.1–100 rad/s at a shear stress of 10 Pa.

2.5.3. Thermal Conductivity

Thermal conductivity was measured at room temperature using the transient plane source (TPS) method on a Hot Disk Thermal Constants Analyzer TPS 500 instrument (Hot Disk AB, Uppsala, Sweden) with a 3.189 mm diameter sensor with Kapton insulation, which was selected based on the dimensions of the specimens. The parameters adjusted included the probing depth, which was set at 4.50 mm, the temperature of the specimen, which was recorded with a portable electronic IR thermometer, and the measurement time and the heating power. To achieve repeatability and reliability of the measurements, a series of five measurements was conducted with an intermediate time of 5 min between measurements.

### 2.5.4. Broadband Dielectric Spectroscopy

To examine the electrical conductivity of the 3D-printed specimens, the BDS technique was chosen, which is ideal for the assessment of bulk material properties. In order to be measured by this technique, 3D-printed specimens required a reduction in thickness from 5 mm to 2 mm through mechanical grinding. For the BDS measurements, Alpha Analyzer frequency response and Quatro Novocontrol automated cooling/heating system were employed. Windeta software (v. 5.87) was employed for recording and processing the results. To perform each measurement, the sample was placed between two circular gold-plated electrodes inside the measurement cell, which was connected via BNC cables to the analyzer. An alternate voltage was applied to the capacitor sample, and the complex dielectric permittivity was measured as a function of frequency, f, in the range of $10^{-1}$ to $10^{-6}$ Hz at ambient temperature conditions. In addition, injection-moulded circular cross-section specimens were also analyzed for comparison with 3D-printed specimens. The electrical conductivity as a function of frequency, $\sigma^*$, was evaluated from the calculated dielectric permittivity, $\varepsilon^*$, according to Equation (1):

$$\sigma^*(\omega) = i \times \omega \times \varepsilon_o \times \varepsilon^*(\omega) \tag{1}$$

where $\omega = 2\pi \times f$ is the angular frequency and $\varepsilon_o$ is the dielectric permittivity of the vacuum.

### 2.5.5. Ohmic Heating Capability

The ohmic behavior of as-produced filament segments was evaluated by measuring their temperature response to changes in applied voltage due to the Joule effect (Figure 2). The experimental setup included an IR camera and a voltage generator with two electrodes, where the voltage and the intensity of the current flow were regulated. The electrodes were placed 1 cm apart. Measurements were taken for fixed voltage values applied for 2 min. Measurements were taken up to the voltage value where the maximum temperature was recorded, resulting in melting of the specimen.

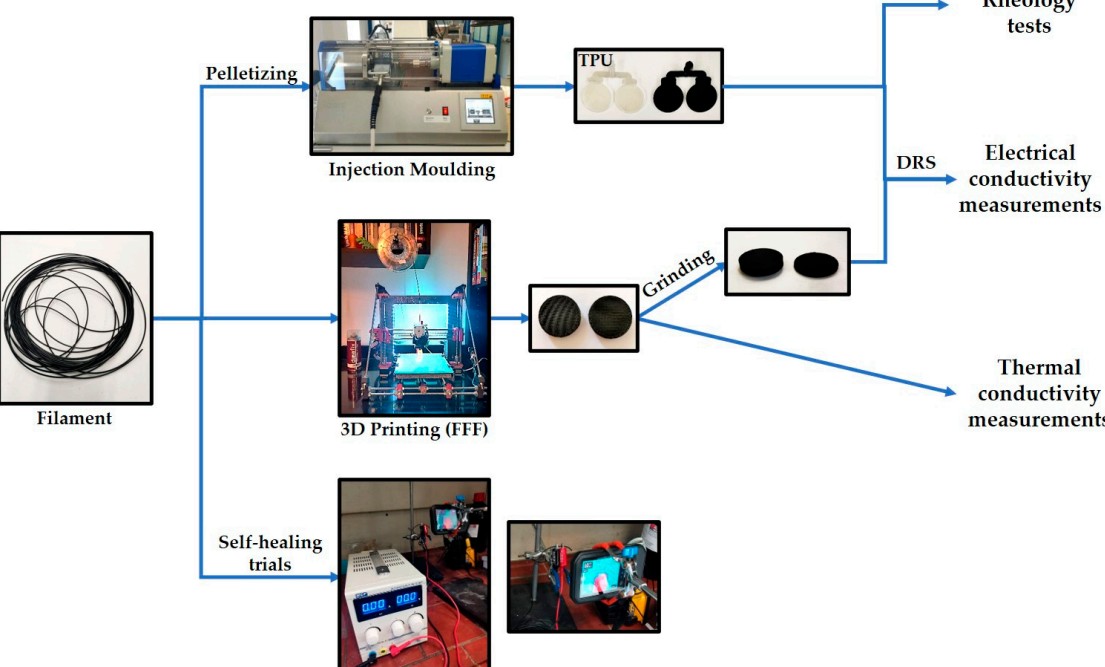

**Figure 2.** Schematic workflow diagram following from the filament to characterization techniques.

## 3. Results

### 3.1. Characterization of Nanomaterials

#### 3.1.1. SEM

The morphology of the entangled MWCNTs_csCVD employed in this study can be seen in Figure 3a,b, where the fibrous form of the synthesized carbon nanotubes was observed. Furthermore, vertically aligned carbon nanotube growth from 35 nm to 65 nm in diameter and ~4 μm in length were achieved (average aspect ratio = 80). From the results of SEM analysis for MWCNTs_cfCVD_bp, illustrated in Figure 3c,d, it can be discerned that the outer diameter of the MWCNTs included in MWCNTs_cfCVD_bp is significantly larger, and in some cases exceeds 100 nm, meaning they may be classified as carbon nanofibers (CNFs). In addition, the aggregates observed may be attributed to either nanotubes or other carbon nanostructures such as amorphous carbon.

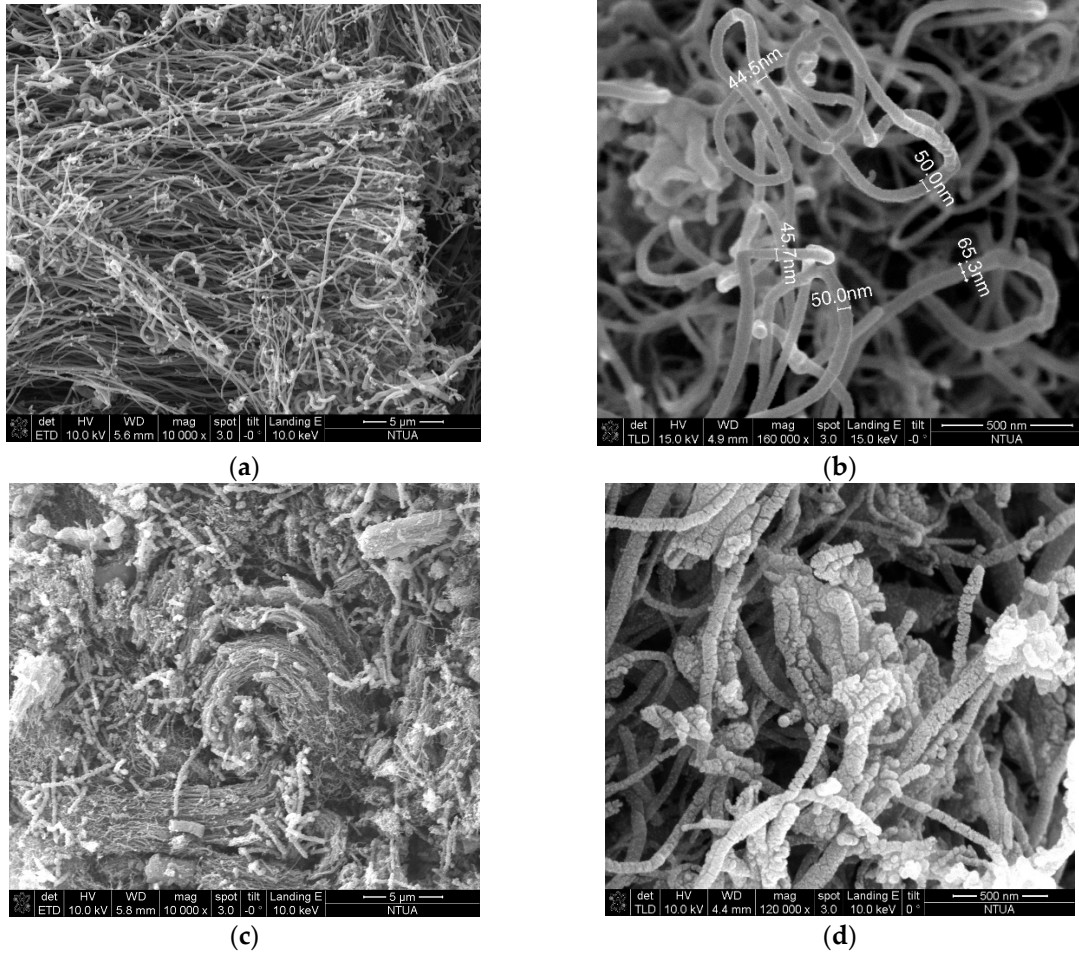

**Figure 3.** SEM micrographs of (**a**,**b**) MWCNTs_csCVD and (**c**,**d**) MWCNTs_cfCVD_bp.

#### 3.1.2. XRD

The XRD spectra of MWCNTs_csCVD (Figure 4a) show the characteristic peaks of the XRD spectra of carbon nanotubes, namely, the large sharp peak for $2\theta = 26°$, corresponding to crystallographic plane (002) diffraction of graphite, and the two united smaller sharp peaks for $2\theta = 43–45°$, reflecting crystallographic planes (100) and (101). In particular, the strong sharp peak for (002) reveals the graphitic structure of MWCNTs [52]. The XRD spectra of MWCNTs_cfCVD_bp (Figure 4b) also shows all the characteristic peaks of the XRD spectra of carbon nanotubes. However, it is worth noting that other smaller peaks at $2\theta = 35°$ and $>50°$ were present, which are likely related to impurities that introduce defect points in crystal lattice formation.

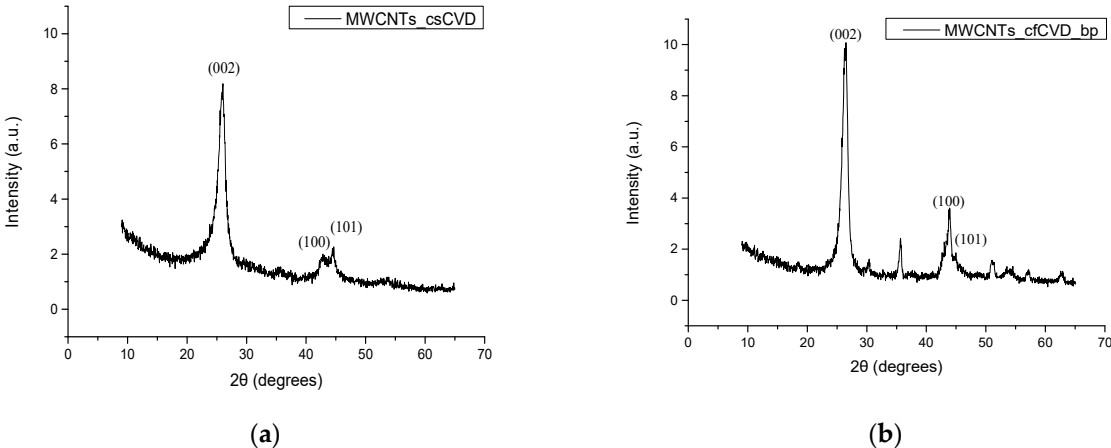

**Figure 4.** XRD of (**a**) MWCNTs_csCVD and (**b**) MWCNTs_cfCVD_bp.

### 3.1.3. Raman Spectroscopy

The Raman spectra of MWCNTs_csCVD (Figure 5a) prove the existence of MWCNTs, as they show the characteristic peaks D (1342 $cm^{-1}$), G (1576 $cm^{-1}$), and G′ (2694 $cm^{-1}$). Generally, the D peak is associated with structural perturbations and defects due to $sp^3$ hybridization, and the G peak is related to the stretched vibrations of $sp^2$ aromatic carbon (C=C) atoms, the degree of graphitization, and the 2D peak [53]. The $I_D/I_G$ ratio represents the percentage of defects and in this case is equal to 0.852. In addition, other peaks are observed after 2600 $cm^{-1}$ which may be due to the iron particles present in the sample. Finally, low-frequency features (<350 $cm^{-1}$) reflecting symmetric radial vibrations (RBM) are not detected in the spectra, which are in direct dependence on the diameter of each nanotube. Due to the absence of the RBM band, it can be concluded that MWCNTs were produced [54].

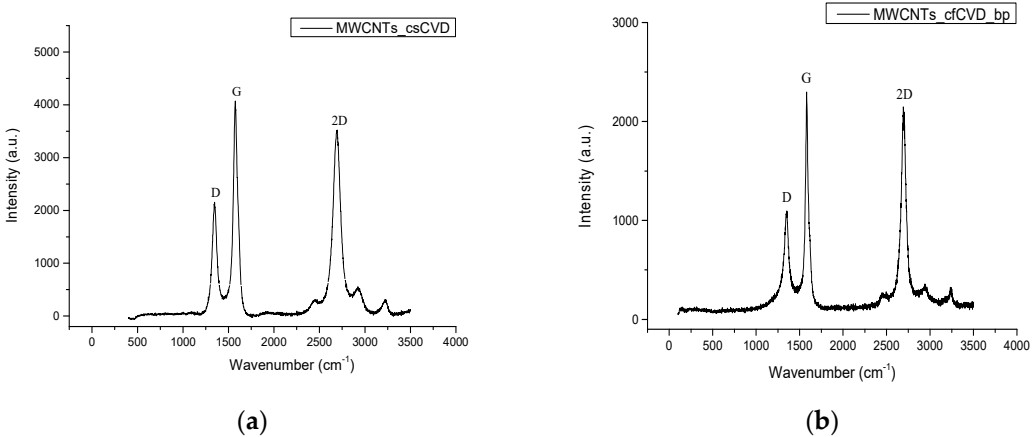

**Figure 5.** Raman spectra of (**a**) MWCNTs_csCVD and (**b**) MWCNTs_cfCVD_bp.

The Raman spectra of MWCNTs_cfCVD_bp (Figure 5b) show the characteristic peaks D, G, and G′ at wavenumbers 1347.64 $cm^{-1}$, 1583.03 $cm^{-1}$, and 2696.49 $cm^{-1}$, respectively. The resulting $I_D/I_G$ ratio is equal to 0.832, which proves that the structures do not have many perturbations. After 2500 $cm^{-1}$, other peaks besides the G′ peak are observed, which are due to impurities contained in the sample, such as amorphous carbon and iron.

### 3.2. Rheology Analysis

Figure 6 shows the frequency (ω) dependence of the storage modulus (G′), loss modulus (G″), and complex viscosity (η*) for 1 wt.% and 10 wt.% prepared samples measured at 240 °C. Consistent values of G′, G″, and η* within the FFF processing window

are desirable for ensuring a uniform rheological response, with high-frequency range values associated with extrusion through the print nozzle and low-frequency range values associated with the bond consolidation process upon deposition [37]. With the increase in the content of each additive, all the studied values are increased, especially at low frequencies, as expected from the restraint on the mobility of the polymer chains and the possible formation of micro-defects introduced due to the interactions between the additives and the polymer matrix [23,55]. As shown in Figure 6a, viscosity increase and shear thinning behavior for 1 wt.% compositions (open symbols) are more pronounced for lab-produced MWCNTs_csCVD, followed by MWCNTs_com, while a moderate shear thinning effect is induced by the commercial rTPU/mTPU_1001_1wt. Shear-thinning behavior is crucial for avoiding shape instability and deformation during the 3D printing process [56,57]. Interestingly, this trend is reverted at the higher nanofiller content of 10 wt.%, where higher viscosity values with steep frequency dependence are observed for mTPU_1001_10wt. In the case of rTPU/MWCNTs_csCVD_10wt nanocomposites, a slight increase in shear thinning response is demonstrated with no significant increase in viscosity in comparison with the 1 wt.% concentration. This effect is considered a favorable characteristic for the development of high filler content nanocomposites for FFF technology, as feedstock materials should be able to be extruded through a small diameter (0.2–0.8 mm) printing nozzle; therefore, high viscosity increases may hinder material feeding and lead to flow instabilities or nozzle blockage. In parallel, higher values of complex viscosity in low-frequency ranges result in printed parts that exhibit better shape retention and mechanical integrity [37]. However, reductions in viscosity offers a counterbalancing effect associated with crossflow upon material deposition, and thus a reduction in voids and air gaps, which improves part structural integrity [58]. Consequently, viscosity should be kept within a medium ideally accompanied by a steep solid-to-liquid transition in order to simultaneously ensure a good level of interlayer adhesion and dimensional stability upon solidification [57,59]. In this context, compositions within the medium viscosity increase range (rTPU/MWCNTs_csCVD 1 & 10 wt& and MWCNTs_cfCVD_bp_10wt), may offer a suitable compromise between processability and structural integrity. Overall, in the majority of nanocomposite samples investigated, a solid-like viscoelastic behavior with $G' \geq G''$ is observed for the selected frequency range. In particular, in the highest contents of additives, it can be observed that $G' \gg G''$, which may be attributed to the liquid-to-solid transition due to the formation of a continuous network of nanofillers in the polymer matrix, impacting the polymer chain relaxation.

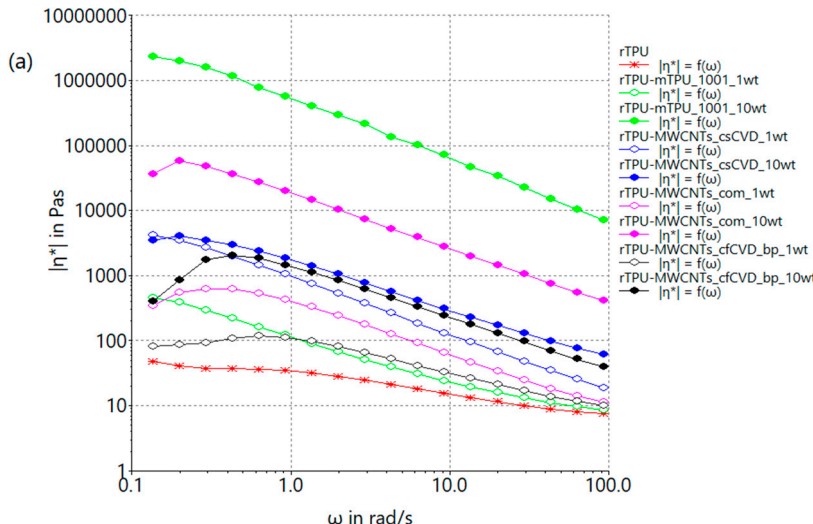

**Figure 6.** *Cont.*

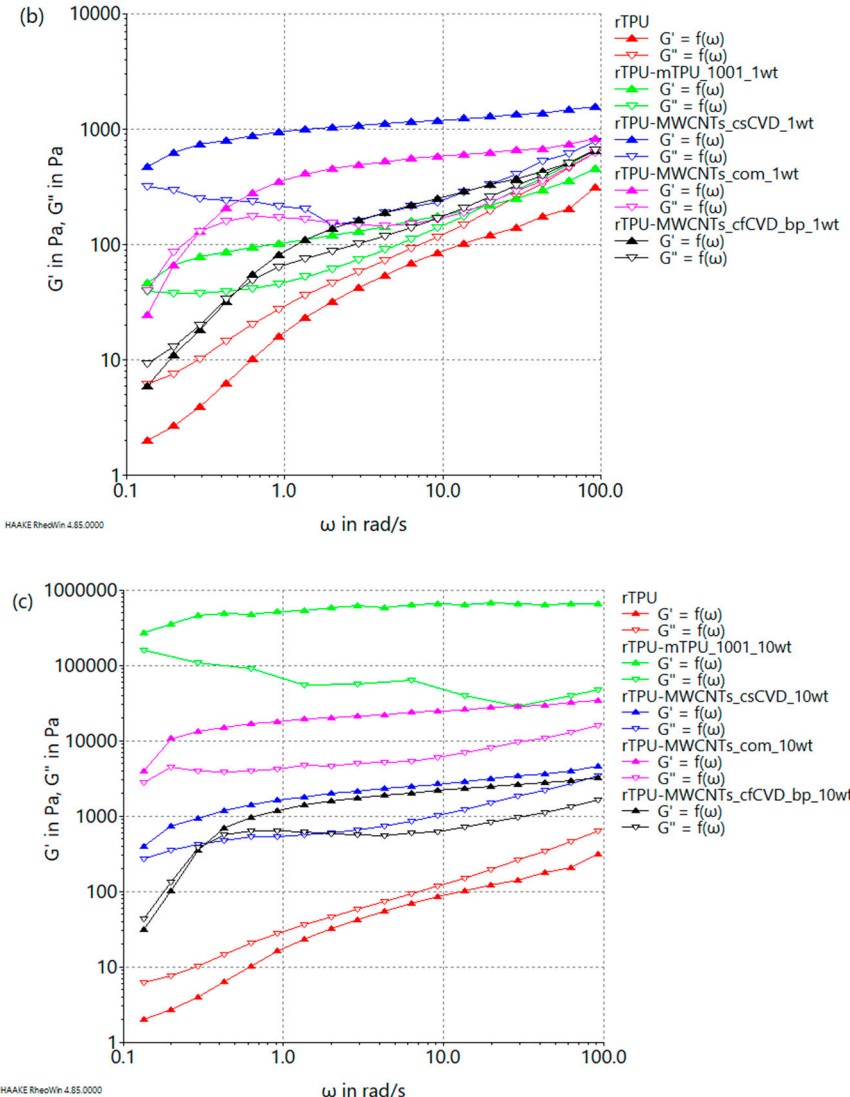

**Figure 6.** Variation in the storage modulus (G′), loss modulus (G″), and complex viscosity (|η*|) as a function of angular frequency (rad/s) at 240 °C. (**a**) Comparative plot of complex viscosity for 1 wt.% and 10 wt.% nanoparticle contents; (**b**) comparative plot of storage and loss modulus for 1 wt.% nanoparticle content; (**c**) comparative plot of storage and loss modulus for 10 wt.% nanoparticle content.

### 3.3. Thermal Conductivity Measurements

As a general trend that can be observed in Figure 7, as the concentration of nanofillers increases, the value of thermal conductivity also increases. Some deviations are observed at intermediate concentrations, which may be associated with the possible degradation of properties from the preceding processes (dispersion, extrusion, 3D printing). The highest thermal conductivity values are observed for rTPU/MWCNTs_csCVD_15wt nanocomposites, where thermal conductivity is more than doubled in comparison with the rTPU matrix. At 10 wt.% concentration, an improvement of thermal conductivity by 79% is demonstrated for mTPU_1001, followed by 56% for lab-synthesized MWCNTs_csCVD and 22% for byproduct MWCNTs_cfCVD_bp, while a non-significant increase is observed for commercial MWCNTs_com. Since the bonding of thermoplastic fibers is thermally operated, tailoring thermal conductivity is considered favorable for the FFF process, as it can allow more efficient heat transfer and material melting, thus improving print quality and reducing printing time [60]. Consequently, mTPU_1001_10wt is promoted by demonstrating the highest improvement of thermal conductivity, while lab-synthesized

MWCNTs_csCVD and byproduct MWCNTs_cfCVD_bp also provide a favorable increase in thermal conductivity in combination with a desirable viscosity range, as discussed in the previous section.

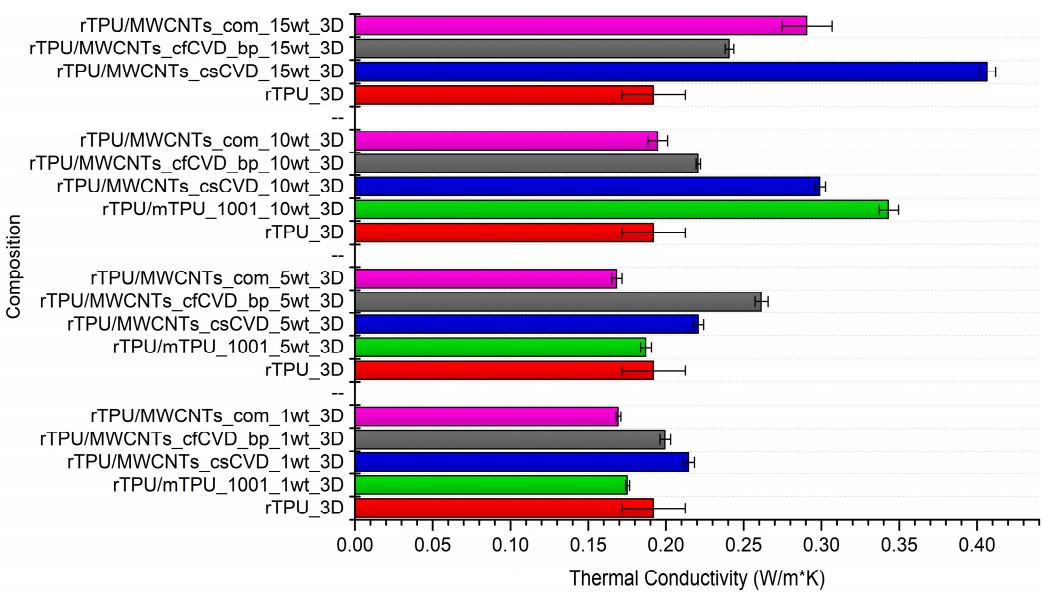

**Figure 7.** Thermal conductivity measurements of the produced 3D-printed nanocomposites. The measurement error is within 3% when the measured sample has a thermal conductivity ranging from 0.005 W/(m K) to 500 W/(m K).

### 3.4. Broadband Dielectric Spectroscopy Results

The electrical conductivity as a function of frequency is presented in Figure 8, where the highest conductivity values were observed for rTPU/mTPU_1001 nanocomposites. Especially in the case of mTPU_1001_10wt, the high electrical and thermal conductivity values obtained indicate that the MWNTs could form well-developed electrical and thermal conductive networks in the TPU matrix. For the majority of the compositions investigated, electrical conductivity at higher frequencies increases almost linearly, which is a typical insulating response. Furthermore, the incorporation of 15 wt.% MWCNTs_com induces an increase in σ* by five orders of magnitude ($6 \times 10^{-7}$ S/cm), while a slight increase of two orders of magnitude is demonstrated after the incorporation of MWCNTs_csCVD ($10^{-12}$ S/cm). The latter values are within the lower regime, thus indicating that carrier conduction via the nanofillers is hindered. Therefore, it can be concluded that a better dispersion of MWCNTs was achieved in the case of rTPU/mTPU_1001 nanocomposites. Through Figure 8b, comparing the injection-moulded and 3D-printed specimens, it was concluded that 3D printing does not introduce process-related artifacts that negatively affect the electrical properties of the materials. From the BDS analysis, even though an improvement was observed in the electric properties of the nanocomposites compared to the pure rTPU, the low conductivity values recorded indicate an insulating behaviour. As demonstrated by other authors for TPU [61–63], with a 5 wt.% content of commercial CNTs, high electrical conductivity values of up to $10^{-3}$–$10^0$ S/cm, respectively, can be obtained. However, it should be mentioned that MWCNTs_csCVD have an aspect ratio two times smaller than MWCNTs_com. In Figure 8, it was noticed that nanocomposites with MWCNTs_csCVD displayed slightly higher values of electrical conductivity in the lowest concentration of 1 wt.% compared to their 5 wt.% and 10 wt.%. counterparts, suggesting that possible nanofiller aggregation may have occurred in higher filler concentrations, which prevented the development of electrical pathways. This also reveals that the percolation threshold is possibly between 1–5 wt.%, as has been demonstrated in other studies [62,63]. The calculated values of electrical conductivity, σ′, at 0.1 Hz, derived from dielectric permittivity are presented in Table 5.

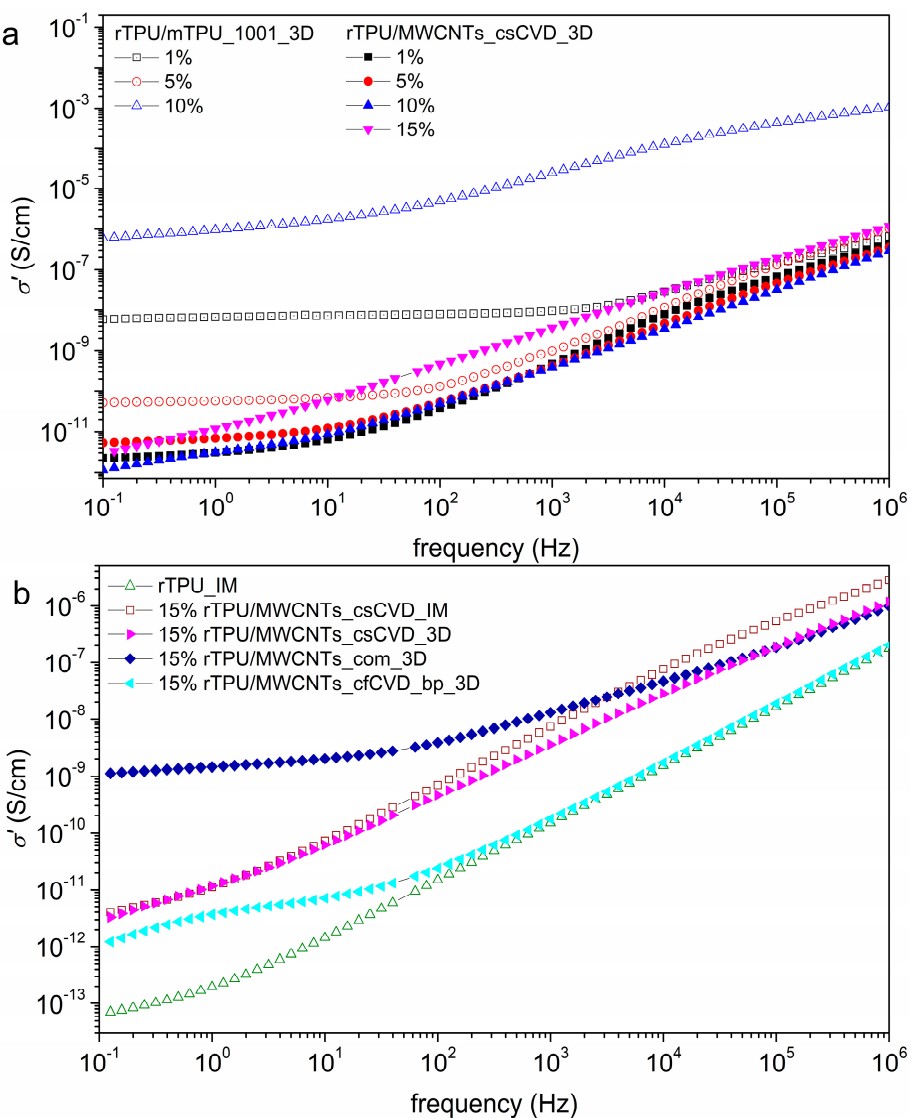

**Figure 8.** Variation in the conductivity (σ′) and as a function of frequency (Hz); (**a**) comparative plot of rTPU/MTPU_1001 and rTPU/MWCNTs_csCVD 3D-printed specimens and (**b**) comparative plot of specimens produced via injection moulding and 3D printing at 15 wt.% nanofiller content.

**Table 5.** Calculated values of electrical conductivity, σ′, at 0.1 Hz, derived from dielectric permittivity.

| Composition | σ′ at 0.1 Hz (S/cm) | Composition | σ′ at 0.1 Hz (S/cm) |
|---|---|---|---|
| rTPU/MWCNTs_csCVD_1wt_3D | $2 \times 10^{-12}$ | rTPU_IM | $7 \times 10^{-14}$ |
| rTPU/MWCNTs_csCVD_5wt_3D | $5 \times 10^{-12}$ | rTPU/MWCNTs_csCVD_15wt_IM | $4 \times 10^{-12}$ |
| rTPU/MWCNTs_csCVD_10wt_3D | $10^{-12}$ | rTPU/MWCNTs_csCVD_15wt_3D | $3 \times 10^{-12}$ |
| rTPU/MWCNTs_csCVD_15wt_3D | $3 \times 10^{-12}$ | rTPU/MWCNTs_com_15wt_3D | $10^{-9}$ |
| rTPU/mTPU_1001_1wt_3D | $6 \times 10^{-9}$ | rTPU/MWCNTs_cfCVD_bp_15wt_3D | $1 \times 10^{-12}$ |
| rTPU/mTPU_1001_5wt_3D | $5 \times 10^{-11}$ | | |
| mTPU_1001_10wt_3D | $6 \times 10^{-7}$ | | |

### 3.5. Ohmic Heating Capability Assessment

After a preliminary screening of the ohmic heating capability of filament compositions listed in Table 2, mTPU_1001_10wt filaments were selected for further analysis after showing promising heating performance. It is noted that the measurement limit of the source was 30–32 V; consequently, filament compositions that might draw current values

above this limit could not be tested with the experimental setup that was employed. The measurement conditions, as well as the results from each set of measurements along with the IR camera images, are provided in Table 6.

**Table 6.** Experimental conditions and results of examining the ohmic heating behavior in mTPU_1001_10wt.

| Applied Voltage (V) | 3.00 | 9.00 | 12.00 | 15.00 |
|---|---|---|---|---|
| Image IR | 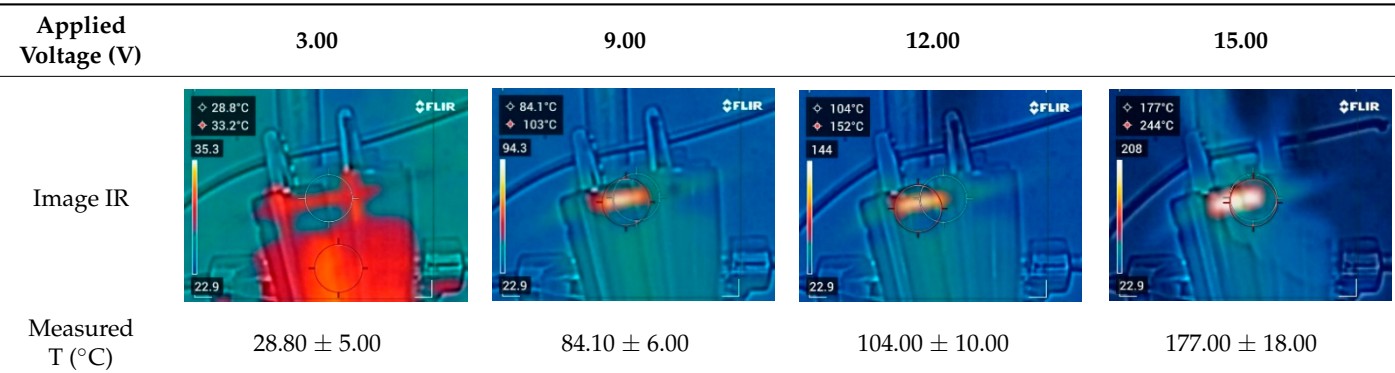 | | | |
| Measured T (°C) | 28.80 ± 5.00 | 84.10 ± 6.00 | 104.00 ± 10.00 | 177.00 ± 18.00 |

As shown in Table 6, it was observed that differential voltage allowed free electron flow through the conducting path of nanofillers, and heating via the Joule effect was shown to result in the conversion/loss of electrical energy in the form of thermal energy. Additionally, it can be observed that the temperature rise presents a linear trend versus the applied voltage, while a similar behavior is also demonstrated in the case of the current versus voltage relationship (Figure 9, with the consequence that nanocomposites can be assumed to follow Ohm's law. By linear correlation, the resistance values of the mTPU_1001_10wt filaments are equal to 101.0 $\Omega$. These resistivity values are consistent for polymer matrix nanocomposites with conductive fillers [64], while the reduced resistivity value of the TPU-based nanocomposite indicates its increased conductivity.

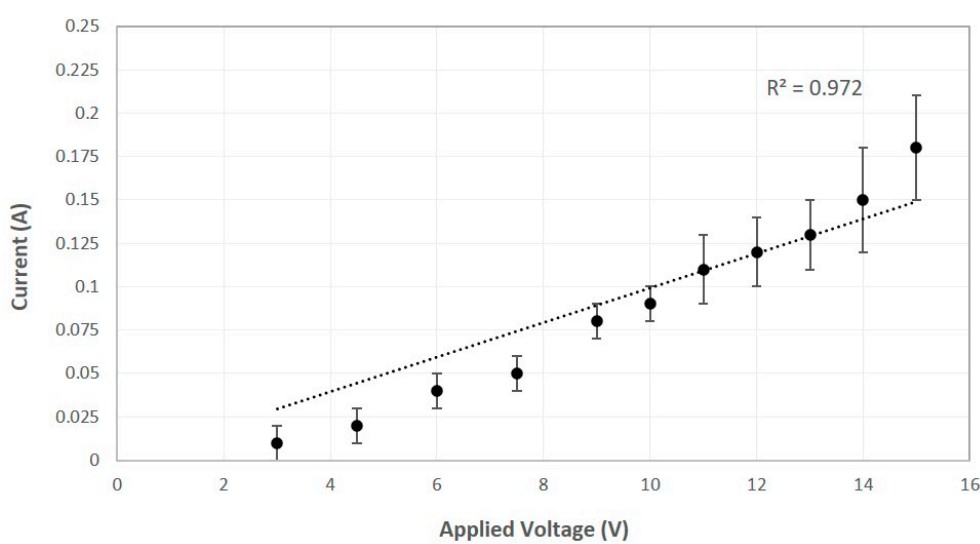

**Figure 9.** Current intensity versus source voltage plots for mTPU_1001_10wt filaments.

To investigate the self-healing ability of the above samples, artificial cracks were introduced on the surface of the sample, perpendicular to the direction of the electric current. SEM analysis took place at the crack location, to be used as a measure of the effectiveness of material remelting due to the Joule effect. The voltage source was set to be equal to 12 V for 10 min, where, within this interval, a current value of 0.12 A and a temperature close to 150 °C was recorded. It should be noted that at higher voltage, the

temperature increased sharply and rapid degradation occurred. The samples were observed in SEM in order to evaluate the morphology of the artificial crack before and after heating with the Joule effect. As shown in Figure 10, in the mTPU_1001_10wt nanocomposites, it was revealed that the artificial cracks were partially restored.

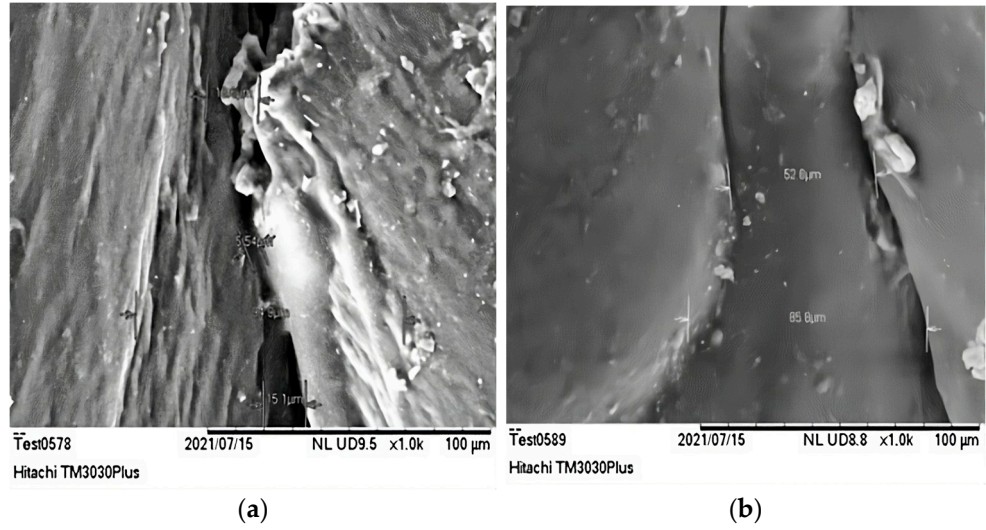

**Figure 10.** SEM micrographs of mTPU_1001_10wt nanocomposite (**a**) in the initial state, (**b**) after heating with the Joule effect.

## 4. Discussion

Large-scale synthesis of nanomaterials, in particular, the production of MWCNTs and MWCNT-byproduct through the CVD method was achieved. Through SEM, XRD, and RAMAN analysis, it was concluded that the MWCNTs with nanoscale diameters produced by the CVD-supported catalyst could contribute significantly to advanced thermal and electrical properties. In contrast, the examined byproducts from the CVD floating catalyst method were tested as promising MWCNT derivatives, but only demonstrated a slight advance in the final properties of the nanocomposites compared to MWCNTs of higher purity; therefore, further purification or chemical modification of the byproducts should be pursued to fully exploit their properties. Considering the effect of the type of additive and the different types of CNTs used, it is noted that the most successful dispersions in terms of the final filaments and the stability of the experimental process occurred for commercial MWCNTs_com, then for MWCNTs_csCVD, and finally for MWCNTs_cfCVD_bp byproduct. This is specifically because of their orientation during mixing, which is mainly affected by their structural characteristics. As demonstrated by the characterization results, MWCNTs_cfCVD_bp filaments were not as effective due to difficulties in dispersion and incorporation into the matrix.

In terms of rheological response, higher values of complex viscosity in low-frequency ranges were observed for all compositions, which is indicative of a better performance in shape retention after material deposition. In parallel, as viscosity reduction provides a counter-balancing effect on reduction in voids and air gaps, compositions within the medium viscosity increase range (rTPU/MWCNTs_csCVD 1 & 10 wt& and MWCNTs_cfCVD _bp_10wt) were identified as a suitable compromise between processability and structural integrity. Overall, in the majority of nanocomposite samples investigated, a solid-like viscoelastictic behavior with $G' \geq G''$ is observed for the selected frequency range, which may be attributed to the liquid-to-solid transition due to the formation of a continuous network of nano-fillers in the polymer matrix, impacting the polymer chain relaxation.

From the thermal conductivity measurements, it is concluded that the highest thermal conductivity values were observed for rTPU/MWCNTs_csCVD_15wt nanocomposites, where thermal conductivity is more than doubled in comparison with the rTPU matrix. In addition, at 10 wt.% concentration, an improvement of thermal conductivity by 56%

was demonstrated for lab-synthesized MWCNTs_csCVD and a 22% improvement was demonstrated for byproduct MWCNTs_cfCVD_bp, while a non-significant increase was observed for commercial MWCNTs_com. Both MWCNTs_csCVD and byproduct MWCNTs_cfCVD_bp provide a favorable increase in thermal conductivity in combination with a desirable viscosity range, thus providing viable options in terms of functionality and processability. It is thus concluded that the synthesized nanomaterials show great promise in tailoring the thermal conductivity of the recycled matrix, which can potentially lead to the improvement of FFF processability and print quality.

Regarding the electrical properties, for the majority of the compositions investigated, electrical conductivity at higher frequencies increased almost linearly, which is a typical insulating response. In the case of mTPU_1001_10wt, the high electrical and thermal conductivity values obtained indicate that the MWNTs could form well-developed electrical and thermal conductive networks in the TPU matrix. Furthermore, the incorporation of 15 wt.% MWCNTs_com induced an increase by five orders of magnitude ($6 \times 10^{-7}$ S/cm), while a slight increase of two orders of magnitude is demonstrated after the incorporation of MWCNTs_csCVD ($10^{-12}$ S/cm). The latter values are within the lower regime, thus indicating that carrier conduction via the nanofillers is hindered. At the same time, comparing the BDS results of specimens produced via 3D printing and injection molding, it was found that the electrical conductivity was not significantly affected by the 3D printing process. On the contrary, its strong correlation with the dispersion and nature of nanomaterials was evident. To address the problem of dispersion, a second polymer phase can be incorporated into the binary polymer composites to further enhance the properties [22] or more additives such as plasticizers or stabilizers can be introduced. Therefore, it can be concluded that further optimization of the MWCNT aspect ratio and filler dispersion should be conducted in order to obtain comparable properties with commercial rTPU/mTPU_1001 nanocomposites.

It was chosen to exploit the Joule effect by applying a sufficient studied electrical stimulus as a trigger for self-healing effects, which is a simple and efficient cost-effective way to repair materials in service. In this study, the self-healing mechanism was activated by temperature, which is the main mechanism that conductive composite materials exhibit [5]. Taking into consideration the different mechanisms of heat transfer from the Joule effect [7,30], the self-healing mechanism that we achieved was dependent on the formation of an electrical percolation network throughout the polymer matrix. In terms of heating efficiency with the Joule effect, artificial cracks were partially restored in mTPU_1001_10wt filaments, which showed an overall improvement in their thermal and electrical properties. The nanocomposites with lab-synthesized CNTs did not reveal self-healing capability under the aforementioned experimental conditions and testing set-up, although they exhibited promising thermal and electrical behavior that can lead to self-healing functionality upon further optimization, leading to the conclusion that with an optimized dispersion process they could display the self-healing mechanism.

## 5. Conclusions

This study demonstrated that recycled and lab-synthesized nanomaterials can be exploited to develop nanocomposites with ohmic heating capability, which can be further exploited towards self-healing applications. Specifically, the nanocomposites based on the recycled rTPU matrix were shown to be able to yield enhanced properties, such as increased thermal and electrical conductivity. In this way, recycled matrix could be a good candidate for self-healing applications while promoting a more sustainable solution. Among the additives investigated, the CNTs produced by the CVD-supported catalyst method showed great promise for inducing improved thermal and electrical properties, comparable to commercial products. However, for self-healing properties, further optimization of their aspect ratio and dispersion in the polymer matrix are required. With a controlled dispersion, they can offer improvements in the thermal and electrical behavior even at low concentrations and exhibit self-healing behavior at lower temperatures. In these circumstances, the mechanical properties of the polymer matrix could be maintained, prolonging the

lifetime of the final materials. Therefore, they can be considered as a cost-effective solution for introducing self-healing capability in 3D-printed specimens, as in their corresponding commercial nanocomposites.

**Author Contributions:** Conceptualization, N.L., A.-F.T. and G.K.; methodology, N.L., E.G., A.-F.T. and G.K.; investigation, N.L., E.G., A.-F.T. and G.K.; resources, C.A.C.; data curation, N.L., E.G., P.A.K. and A.K.; writing—original draft preparation, N.L. and A.-F.T.; writing—review and editing, G.K. and E.G.; supervision, A.-F.T. and C.A.C.; project administration, C.A.C.; funding acquisition, C.A.C. All authors have read and agreed to the published version of the manuscript.

**Funding:** This work is funded by the European Union's Horizon 2020 Research and Innovation program, entitled: 'Recycling and Repurposing of Plastic Waste for Advanced 3D Printing Applications' (Repair3D) under GA No. 814588.

**Institutional Review Board Statement:** Not applicable.

**Informed Consent Statement:** Not applicable.

**Data Availability Statement:** The data presented in this study are available on request from the corresponding author. The data are not publicly available due to the confidentiality of the Repair3D project.

**Acknowledgments:** The authors would like to thank Konstantinos Zafeiris and Dionisis Semitekolos for their assistance on the rheological and thermal conductivity measurements, respectively. The recycled polymers were provided by Calzaturificio Dalbello S.r.l.

**Conflicts of Interest:** The authors declare no conflict of interest. The funders had no role in the design of the study; in the collection, analyses, or interpretation of data; in the writing of the manuscript, or in the decision to publish the results.

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
