# Peer review of "Development of CNT-Based Nanocomposites with Ohmic Heating Capability towards Self-Healing Applications in Extrusion-Based 3D Printing Technologies"

_carbon, 2023_

Round 1

Reviewer 1 Report

Comments and Suggestions for Authors

1.    The manuscript is missing a lot of relevant literature references. A non-exhaustive list of such sources is provided below. The authors should discuss them in the manuscript not only to present the current state-of-the-art but also to compare their results with respect to previous research.

a.     https://doi.org/10.1016/j.sna.2017.07.020

b.     https://doi.org/10.1021/acs.iecr.3c01409

c.     https://doi.org/10.1016/j.compositesb.2019.107600

2.     Please add the following references in the rheology section:

a.     https://doi.org/10.1021/acsapm.0c01228

b.     https://doi.org/10.1016/j.progpolymsci.2019.101162

c.     https://doi.org/10.1016/j.progpolymsci.2021.101411

3.     An additional criticism is that much of the manuscript simply states what has been found from the experiments rather than synthesizing the results to provide clear insights into the underlying physics/chemistry that govern the observations. Much of the manuscript reads as a paragraph-by-paragraph listing of the results obtained from printing the filled polymer but with the absence of narrative tying them together.

4.     What was the accelerating voltage in the SEM and under what vacuum mode was it operated?

5.     Please improve the quality of Figures 6 and 8; it is difficult to see the legend clearly.

Comments on the Quality of English Language

Minor editing of the grammar and spelling check are required.

Author Response

We highly appreciate the time and effort that the reviewers dedicated to providing feedback on our manuscript and are grateful for the insightful comments received. We have incorporated most of the suggestions made by the reviewers. Those changes are highlighted within the attached manuscript in tracked changes mode. Please see below, for a point-by-point response to the reviewers’ comments and concerns. All section/page numbers refer to the revised manuscript file.

A1. The manuscript is missing a lot of relevant literature references. A non-exhaustive list of such sources is provided below. The authors should discuss them in the manuscript not only to present the current state-of-the-art but also to compare their results with respect to previous research.

  1. https://doi.org/10.1016/j.sna.2017.07.020
  2. https://doi.org/10.1021/acs.iecr.3c01409
  3. https://doi.org/10.1016/j.compositesb.2019.107600

Answer: Thank you for pointing this out. Literature review included in the Introduction and Discussion sections has been further enriched with additional references from previous studies.

p.3 “Additionally, the combination of 3D printing technologies with material feed-stocks with embedded functionalities is considered highly promising for the production of functional parts with extended service life [41]. Several researchers have shown that it is possible to fabricate 3D printed sensors by using functionalized nanocomposites based on CNTs and TPU [42-44]. Niu, Yang et al. [45] showed that 3D printed CNT/TPU composites could offer both EMI shielding performance and mechanical properties in next generation electronic devices. Wu, Zou et al. [46] proves the possibility of 3D printing self-healable strain gauges and humidity sensors from CS/CNT nanocomposite because of their high stretchability and conductivity.”

Results comparison is conducted in terms of CNTs-based self-healing polymer composites [28-32], electrical percolation threshold [61-63], resistivity values for polymer matrix nanocomposites with conductive fillers [64], however due to the novelty of materials used (recycled TPU, in-house produced nanomaterials and byproducts of CVD process), further comparison and benchmark properties were challenging to establish.

A2. Please add the following references in the rheology section:

  1. https://doi.org/10.1021/acsapm.0c01228
  2. https://doi.org/10.1016/j.progpolymsci.2019.101162
  3. https://doi.org/10.1016/j.progpolymsci.2021.101411

Answer: Thank you for this suggestion. Descriptions and references in Section 3.2 have been updated accordingly, as summarised below:

p.10 “Consistent values of G′, G″ and η* within the FFF processing window are desirable for ensuring a uniform rheological response, with high-frequency range values associated with the extrusion through the print nozzle and low-frequency range associated with the bond consolidation process upon deposition [37].”

p.10 “With the increase in the content of each additive, all the studied values are increased, especially ats low frequencies, as expected from the restraint on the mobility of polymer chains and possible formation of micro-defects introduced due to the interactions between additives and the polymer matrix [55, 23].”

p.10 “In parallel, higher values of complex viscosity in low-frequency range result in printed parts that exhibit better shape retention and mechanical integrity [37]. However, reduction of viscosity offers a counter-balancing effect associated with crossflow upon material deposition, and thus reduction of voids and air gaps, which improves part structural integrity [58].”

p.10 “In this context, compositions within the medium viscosity increase range (rTPU/MWCNTs_csCVD 1 & 10 wt& and MWCNTs_cfCVD_bp_10wt), may offer a suitable compromise between processability and structural integrity.”

A3. An additional criticism is that much of the manuscript simply states what has been found from the experiments rather than synthesizing the results to provide clear insights into the underlying physics/chemistry that govern the observations. Much of the manuscript reads as a paragraph-by-paragraph listing of the results obtained from printing the filled polymer but with the absence of narrative tying them together.

Answer: Thank you for pointing this out. The following descriptions have been added to provide further insights on results obtained:

p.10 “In this context, compositions within the medium viscosity increase range (rTPU/MWCNTs_csCVD 1 & 10 wt. & and MWCNTs_cfCVD_bp_10wt), may offer a suitable compromise between processability and structural integrity.”

p.11 “Consequently, mTPU_1001_10wt is promoted by demonstrating the highest improvement of thermal conductivity, while lab-synthesized MWCNTs_csCVD and byproduct MWCNTs_cfCVD_bp also provide a favorable increase of thermal conductivity, in combination with a desirable viscosity range, as discussed in the previous section.”

p.12 “Especially in the case of mTPU_1001_10wt, the high electrical and thermal conductivity values obtained indicate that the MWNTs could form well-developed electrical and thermal conductive networks in the TPU matrix.”

A4.   What was the accelerating voltage in the SEM and under what vacuum mode was it operated?

Answer: Hitachi TM3030 tabletop microscope (QUANTAX 70) was employed for Scanning Electron Microscopy (SEM), equipped with a diaphragm pump and a turbomolecular pump, and operated in secondary electrons mode with an acceleration voltage of 10 and 15 keV at a vacuum of 10-5 Pa in order to analyse the morphology of the synthesized nanomaterials. Section 2.5.1 has been updated accordingly (p.6).

A5.   Please improve the quality of Figures 6 and 8; it is difficult to see the legend clearly.

Answer: Figures 6, 7 and 8 have been revised to improve clarity.

In addition to the above comments, a few spelling and grammatical errors have been corrected. We look forward to hearing from you regarding our submission and to respond to any further questions and comments you may have.

Reviewer 2 Report

Comments and Suggestions for Authors

The manuscript authored by Charitidis et al. presents a significant contribution in the field of nanocomposite materials, particularly in the context of their application in Extrusion-based 3D Printing Technologies for self-healing purposes. This work focuses on the development of carbon nanotube (CNT)-based nanocomposites capable of ohmic heating, utilizing a recycled thermoplastic polyurethane (TPU) matrix and multi-walled carbon nanotubes (MWCNTs) as fillers. The resulting nanocomposites were comprehensively examined for their mechanical and electrical properties. I highly recommend the manuscript for publication in the journal Carbon, although a few key concerns need to be addressed to further enhance the quality of the work:

Figure Clarity: Figures 6, 7, and 8 present valuable data; however, their legibility needs improvement. To enhance reader comprehension, please consider revising these figures with larger fonts and improved clarity.

Post-Healing Mechanical Analysis: It would be beneficial to conduct mechanical analysis on the nanocomposites after the self-healing process. This post-healing mechanical characterization can provide insights into the effectiveness and durability of the self-healing mechanism.

Thermal Properties: Including thermal properties such as Thermogravimetric Analysis (TGA) and Differential Scanning Calorimetry (DSC) data would significantly strengthen the manuscript. Understanding the thermal behavior of the nanocomposites is crucial for their practical applications.

Statistical Data: To enhance the reliability of the presented results, it is essential to provide standard deviation and error bars, especially when discussing mechanical and electrical properties. This will allow readers to assess the data's reproducibility and statistical significance.

Literature Comparison and Self-Healing Mechanism: To better contextualize the findings, consider comparing the obtained results with relevant literature values for mechanical properties. Furthermore, provide a detailed explanation of the underlying mechanisms involved in the self-healing process. This will enrich the discussion and improve the manuscript's overall quality.

Conclusion Section: The conclusion section needs improvement. A strong and succinct conclusion is vital to summarize the key findings and their implications.

Comments on the Quality of English Language

The manuscript authored by Charitidis et al. presents a significant contribution in the field of nanocomposite materials, particularly in the context of their application in Extrusion-based 3D Printing Technologies for self-healing purposes. This work focuses on the development of carbon nanotube (CNT)-based nanocomposites capable of ohmic heating, utilizing a recycled thermoplastic polyurethane (TPU) matrix and multi-walled carbon nanotubes (MWCNTs) as fillers. The resulting nanocomposites were comprehensively examined for their mechanical and electrical properties. I highly recommend the manuscript for publication in the journal Carbon, although a few key concerns need to be addressed to further enhance the quality of the work:

Figure Clarity: Figures 6, 7, and 8 present valuable data; however, their legibility needs improvement. To enhance reader comprehension, please consider revising these figures with larger fonts and improved clarity.

Post-Healing Mechanical Analysis: It would be beneficial to conduct mechanical analysis on the nanocomposites after the self-healing process. This post-healing mechanical characterization can provide insights into the effectiveness and durability of the self-healing mechanism.

Thermal Properties: Including thermal properties such as Thermogravimetric Analysis (TGA) and Differential Scanning Calorimetry (DSC) data would significantly strengthen the manuscript. Understanding the thermal behavior of the nanocomposites is crucial for their practical applications.

Statistical Data: To enhance the reliability of the presented results, it is essential to provide standard deviation and error bars, especially when discussing mechanical and electrical properties. This will allow readers to assess the data's reproducibility and statistical significance.

Literature Comparison and Self-Healing Mechanism: To better contextualize the findings, consider comparing the obtained results with relevant literature values for mechanical properties. Furthermore, provide a detailed explanation of the underlying mechanisms involved in the self-healing process. This will enrich the discussion and improve the manuscript's overall quality.

Conclusion Section: The conclusion section needs improvement. A strong and succinct conclusion is vital to summarize the key findings and their implications.

Author Response

We highly appreciate the time and effort that the reviewers dedicated to providing feedback on our manuscript and are grateful for the insightful comments received. We have incorporated most of the suggestions made by the reviewers. Those changes are highlighted within the attached manuscript in tracked changes mode. Please see below, for a point-by-point response to the reviewers’ comments and concerns. All section/page numbers refer to the revised manuscript file.

Comments and Suggestions for Authors

The manuscript authored by Charitidis et al. presents a significant contribution in the field of nanocomposite materials, particularly in the context of their application in Extrusion-based 3D Printing Technologies for self-healing purposes. This work focuses on the development of carbon nanotube (CNT)-based nanocomposites capable of ohmic heating, utilizing a recycled thermoplastic polyurethane (TPU) matrix and multi-walled carbon nanotubes (MWCNTs) as fillers. The resulting nanocomposites were comprehensively examined for their mechanical and electrical properties. I highly recommend the manuscript for publication in the journal Carbon, although a few key concerns need to be addressed to further enhance the quality of the work:

B1. Figure Clarity: Figures 6, 7, and 8 present valuable data; however, their legibility needs improvement. To enhance reader comprehension, please consider revising these figures with larger fonts and improved clarity.

Answer: Figures 6, 7 and 8 have been revised to improve clarity.

B2. Post-Healing Mechanical Analysis: It would be beneficial to conduct mechanical analysis on the nanocomposites after the self-healing process. This post-healing mechanical characterization can provide insights into the effectiveness and durability of the self-healing mechanism.

Answer: The suggestion to conduct post-healing mechanical analysis is indeed an interesting aspect that we appreciate. Our primary focus in this study was on the development of self-healing materials from recycled resources and the initial assessment of their self-healing capabilities, particularly with regard to superficial damage (surface cracks). We aimed to establish the feasibility and effectiveness of the self-healing mechanism in different material compositions. As such, we did not include post-healing mechanical analysis in the current study due to resource and scope limitations. In our future research work, we are planning to investigate post-healing mechanical properties using the most promising material compositions identified in this study. This analysis will allow us to gain deeper insights into the performance and robustness of the self-healing mechanism under real-world conditions.

B3. Thermal Properties: Including thermal properties such as Thermogravimetric Analysis (TGA) and Differential Scanning Calorimetry (DSC) data would significantly strengthen the manuscript. Understanding the thermal behaviour of the nanocomposites is crucial for their practical applications.

Answer: Thank you for this suggestion. Due to the extend of experimental work already included in the present study, it was decided to focus on thermal conductivity in terms of thermal properties assessment, due to its contribution to convective transfer of heat generated by the Joule effect. Currently there is another work prepared by our group on this topic (thermal transitions via thermo-analytical techniques at different processing stages), and thus this analysis will be addressed separately, to avoid potential conflicts.

B4. Statistical Data: To enhance the reliability of the presented results, it is essential to provide standard deviation and error bars, especially when discussing mechanical and electrical properties. This will allow readers to assess the data's reproducibility and statistical significance.

Answer: Thank you for pointing this out. Error bars have been included where applicable, i.e., in Figure 7, as the standard deviation of 5 measurements conducted. To examine the electrical conductivity of the 3D printed specimens, the BDS technique was chosen, which is ideal and highly precise for the assessment of bulk material properties. As the electrical conductivity as a function of frequency, σ*, was evaluated from the calculated dielectric permittivity, ε*, conductivity values are typically reported for a specific frequency value (in our case 0.1 Hz), for the comparative assessment of different compositions.

B5. Literature Comparison and Self-Healing Mechanism: To better contextualize the findings, consider comparing the obtained results with relevant literature values for mechanical properties. Furthermore, provide a detailed explanation of the underlying mechanisms involved in the self-healing process. This will enrich the discussion and improve the manuscript's overall quality.

Answer: Thank you for this suggestion. Since the current investigation is focusing on the evaluation of rheological properties, thermal conductivity, dielectric response, resistivity and ohmic heating capability, results comparison is conducted in terms of CNTs-based self-healing polymer composites [28-32], electrical percolation threshold [61-63], resistivity values for polymer matrix nanocomposites with conductive fillers [64]. Due to the novelty of materials used (recycled TPU, in-house produced nanomaterials and byproducts of CVD process), further comparison and benchmark properties were challenging to establish. In addition, the underlying mechanisms in the self-healing process have been further elaborated in the Discussion section:

p.18 “In this study, the self-healing mechanism was activated by temperature, which is the main mechanism that conductive composite materials exhibit [5]. Taking into consideration the different mechanisms of heat transfer from the Joule effect [7, 65], the self-healing mechanism that we achieved was dependent on formation of an electrical percolation network throughout the polymer matrix. In terms of heating efficiency with the Joule effect, artificial cracks have been partially restored in mTPU_1001_10wt filaments, which showed an overall improvement in their thermal and electrical properties.”

B6. Conclusion Section: The conclusion section needs improvement. A strong and succinct conclusion is vital to summarize the key findings and their implications.

Answer: Thank you for pointing this out. Descriptions in Discussion and Conclusions sections have been further elaborated to summarize key findings and their implications, with some indicative (non-exhaustive) descriptions listed below:

p.17: In terms of rheological response, higher values of complex viscosity in low-frequency range were observed for all compositions, which is indicative of a better performance in shape retention after material deposition. In parallel, as viscosity re-duction provides a counter-balancing effect on reduction of voids and air gaps, com-positions within the medium viscosity increase range (rTPU/MWCNTs_csCVD 1 & 10 wt& and MWCNTs_cfCVD_bp_10wt) were identified as a suitable compromise be-tween processability and structural integrity. Overall, in the majority of nanocompo-site samples investigated, a solid-like viscoelastictic behavior with G’≥G” is observed for the selected frequency range, which may be attributed to the liquid-to-solid transi-tion due to the formation of continuous network of nano-fillers in the polymer matrix, impacting the polymer chain relaxation.

p.17-18: Both MWCNTs_csCVD and byproduct MWCNTs_cfCVD_bp provide a favorable in-crease of thermal conductivity, in combination with a desirable viscosity range, thus providing viable options in terms of functionality and processability.

p.18: In the case of mTPU_1001_10wt, the high electrical and thermal conductivity values obtained indicate that the MWNTs could form well-developed electrical and thermal conductive networks in the TPU matrix.

p.18: In this way, recycled matrix could be a good candidate for self-healing applications while promoting a more sustainable solution.

p.18: With a controlled dispersion, they can offer improvements in the thermal and electrical behavior even at low concentrations and exhibit self-healing behavior in lower temperatures. In these circumstances, the mechanical properties of the polymer matrix could be maintained, prolonging the lifetime of the final materials.

In addition to the above comments, a few spelling and grammatical errors have been corrected. We look forward to hearing from you regarding our submission and to respond to any further questions and comments you may have.

Round 2

Reviewer 2 Report

Comments and Suggestions for Authors

The manuscript has undergone a thorough revision, meeting the necessary criteria for acceptance in its current state.